

# Transcriptome analysis of flavonoid biosynthesis in safflower flowers grown under different light intensities

Chaoxiang Ren[1,2], Jie Wang[1,2], Bin Xian[1,2], Xiaohui Tang[1,2],
Xuyun Liu[3], Xueli Hu[3], Zunhong Hu[3], Yiyun Wu[1,2], Cuiping Chen[1,2],
Qinghua Wu[1,2], Jiang Chen[1,2] and Jin Pei[1,2]

[1] College of Pharmacy, Chengdu University of Traditional Chinese Medicine, Chengdu, China
[2] Development and Utilization of Chinese Medicine Resources, State Key Laboratory Breeding
Base of Systematic Research, Chengdu, China
[3] Industrial Crop Institute of Yunnan Academy of Agricultural Sciences, Yunnan, China

## ABSTRACT

**Background:** Safflower (*Carthamus tinctorius* L.) is a domesticated species with a long history of cultivation and widespread distribution across the globe, and light plays an important role in controlling its distribution boundary. Flowers from safflower have been widely used in traditional Chinese medicine because of their ability to improve cerebral blood flow. Flavonoids are the main active compounds in safflower and have many pharmacological effects. In this study, we aimed to explore the relationship between different light intensities and flavonoid biosynthesis in safflower flowers cultivated in greenhouse.

**Methods:** The transcriptome of safflower flowers grown under different light intensities were sequenced through BGISEQ-500 platform. After assembled and filtered, Unigenes were annotated by aligning with seven functional databases. Differential expression analysis of two samples was performed with the DEseq2 package. Differentially expressed genes (DEGs) related with flavonoids biosynthesis were analyzed by Real-time PCR (RT-PCR). Flavonoids accumulation in flowers were determined by high performance liquid chromatography and spectrophotometer.

**Results:** Transcriptome analysis of safflower flowers cultivated under different light intensities was performed. A total of 99.16 Gb data were obtained, and 78,179 Unigenes were annotated. Among the DEGs, 13 genes were related to flavonoid biosynthesis. The differential expressions of seven key genes were confirmed by RT-PCR. In addition, the levels of some flavonoids were measured in safflower flowers grown under different light intensities. *CtHCT3* gene expression showed a significantly negative correlation with kaempferol content in safflower grown under different light intensities.

**Conclusion:** Our results strongly suggested that the reduction in light intensity in a suitable range promoted flavonoid biosynthesis in safflower flowers. We suggest that the expressions of *HCT* genes played an important role in flavonoid accumulation in safflower flowers. Our study lays a foundation for further research on the effects of light on flavonoid biosynthesis in safflower.

Corresponding authors
Jiang Chen, janshen1986@163.com
Jin Pei, peixjin@163.com

# INTRODUCTION

Safflower, *Carthamus tinctorius* L. ($2n = 2x = 24$), is a member of the family Asteraceae. It is an annual and a predominantly self-pollinated species. Safflower is a domesticated species with a long history of cultivation and widespread distribution across the world (*Dempewolf, Rieseberg & Cronk, 2008*). With the introduction of inexpensive synthetic dyes in the middle of the last century, the importance of its flowers as a source of dye has almost vanished. However, the breeding of safflower for its high oil content and the modified fatty acid composition in its seeds (*Knowles, 1972*) has given rise to new varieties that led to its re-establishment as an important high-quality oil crop for cooking and industrial purposes in many parts of the world (*Ashri et al., 1977*). On the basis of the morphological variability that exists in *Carthamus tinctorius* L., researchers proposed seven 'centers of similarity' (the Far East, India–Pakistan, the Middle East, Egypt, Sudan, Ethiopia and Europe) that have predominant morphotypes at each center (*Knowles, 1969*). However, China, unlike other countries, has been using its flowers as a medicinal herb for nearly 2,000 years and has already become a special cultivation center. Previous studies on the effect of environmental factors on safflower distribution, showed that light was the most important factor explaining safflower distribution boundaries (*Wu, 2017*).

Plants have adopted the ability to sense light signals which is one of the most important environmental factors for plants growth, including light quantity (intensity), quality (wavelength), direction and duration. The response of plants to light occurs in various developmental processes, such as seedling photomorphogenesis, circadian rhythms, flower induction as well as the accumulation of secondary metabolites (*Jiao, Lau & Deng, 2007*). Numerous studies have shown that light intensity can influence the biosynthesis of flavonoids in other species. For instance, shade had a significantly negative effect on contents of total flavonoid, kaempferol, quercetin and isoquercitrin in leaves of *Cyclocarya paliurus*. However, the greatest accumulation of total flavonoid in the leaves was observed in intermediate shade treatment (*Deng et al., 2012*). In *Epimedium pseudowushanense* B.L. Guo, the flavonoid contents varied with five different light intensity levels (light intensity was getting stronger from level 1 to level 5) and the largest amount of epimedin C was produced at light intensity level 4 (*Pan et al., 2017*). In *Begonia semperflorens*, high light stress promoted anthocyanin synthesis in the leaves (*Wang et al., 2018*).

Flavonoids are ubiquitous secondary metabolites that have various functions in plant physiology and ecology (*Tian, Pang & Dixon, 2008*). At present, the basic metabolic pathway of flavonoid biosynthesis is well known, especially in *Arabidopsis thaliana* (*Hai, Huang & Tang, 2010*; *Saito et al., 2013*). *P*-coumaroyl CoA is the substrate of two enzymes at the junction of the metabolic routes leading to flavonoids or to phenylpropanoid compounds (Fig. 1). One of enzymes is shikimate O-hydroxycinnamoyl transferase (HCT) (EC:2.3.1.133), which leads to biosynthesis of two major lignin building

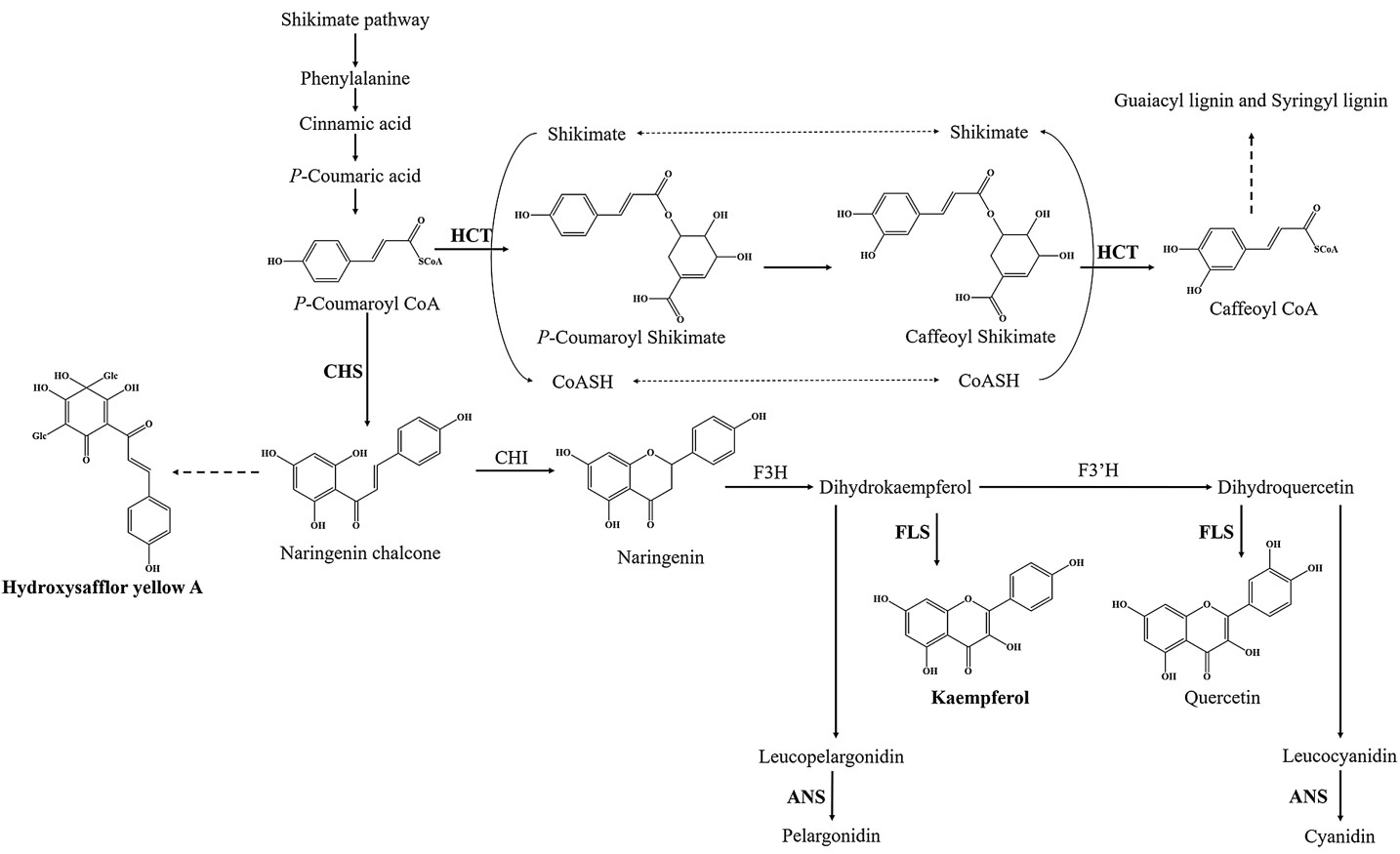

**Figure 1 Proposed a part of flavonoid biosynthesis pathway in safflower.** HCT, Shikimate O-hydroxycinnamoyl transferase; CHS, Chalcone synthase; FLS, Flavonol synthase; ANS, Anthocyanidin synthase.

units, namely, the guaiacyl and syringyl units (*Besseau et al., 2007*). Another is chalcone synthase (CHS), which catalyzes *p*-coumaroyl CoA and three malonyl CoA molecules to form the naringenin chalcone, and naringenin chalcone subsequently is converted to naringenin by chalcone isomerase (CHI). Flavanone 3-hydroxylase (F3H), flavonol synthase (FLS) (EC:1.14.20.6) and (F3'H) catalyze the formation of many kinds of flavonol, such as kaempferol and quercetin. Downstream, anthocyanidin synthase (ANS) (EC:1.14.20.4) catalyzes the formation of anthocyanin, such as pelargonidin and cyanidin. However, only a few genes have been identified in flavonoid biosynthesis pathway of safflower, such as *CHI* and *F3H* (*Guo et al., 2019*; *Ren et al., 2019*; *Tu et al., 2016*). Therefore, transcriptome analysis is one of the best approaches to find functional genes and to reveal the flavonoid biosynthesis in safflower.

Additionally, pharmacological studies have shown that flavonoids in safflower have several pharmacological effects including cardioprotective (*Jin et al., 2008*), vasodilative effects and exhibit anti-hypertensive (*Asgarpanah & Kazemivash, 2013*), anti-coagulation and anti-thrombotic (*Hong et al., 2015*) properties. Specially, hydroxysafflor yellow A (HSYA) is the most widely studied chemical in safflower. Recent research reports showed that HSYA had anti-inflammatory activity and a protective effect on ischaemic cardiac

dysfunction, and it was able to weaken coronary heart disease and enhance the rate of wound closure (*Gao et al., 2018*; *Zhou et al., 2018*; *Zou et al., 2018*). Besides, kaempferol is a natural flavonol present in different plant species that possesses potent anti-inflammatory properties (*Devi et al., 2015*). Kaempferol, as an index compound of safflower, is listed along with HSYA in the *Chinese Pharmacopoeia Commission (2015)*.

To reveal the relationship between light intensity and flavonoid biosynthesis in safflower, transcriptomes of safflower flowers grown under different light intensities were sequenced in this study. Different genes related to flavonoid biosynthesis were analyzed. In addition, HSYA, kaempferol and total flavonoid contents in the safflower flowers were determined. The correlation between gene expression and the flavonoid content was analyzed. Our results provide a basis for further study of the influence of light intensity on flavonoid biosynthesis in safflower flowers.

## MATERIALS AND METHODS

### Plant materials

The safflower flowers used in this study were obtained from the phytotron at the Chengdu University of Traditional Chinese Medicine in Chengdu, China. Prior to the experiment, the plants were grown at 25 °C during the day and at 18 °C at night; the relative humidity ranged from 50% to 60%. Safflower plants with the same growth status were selected and grown in a peat: vermiculite: perlite (3:1:1) mixture in a phytotron room. The 8 months old plants were divided into three groups (30 independent samples in each group) with equal size buds. Plants were illuminated with light-emitting diodes (LEDs) among three groups. The high light (HL) group was grown under 40,000 Lux. The second group, middle light (ML), was grown under 20,000 Lux. The third group, low light (LL), was grown under 10,000 Lux. Light intensity settings referred to light compensation point (LCP) and light saturation point (LSP) of safflower (*Guan et al., 2007*). Three different treatments lasted from budding to blooming. Blooming flowers were collected per group from the same location on the plants. These samples were immediately frozen at −80 °C. Three biological replicates were used in each treatment.

### RNA extraction

Flowers were ground in liquid nitrogen, and the total RNA was isolated by TRIzol reagent (Invitrogen, Carlsbad, CA, USA). To remove DNA, an aliquot of total RNA was treated with DNase (Takara, Dalian, China) by using the standard protocol described by the manufacturer. The purity of the RNA samples according to the A260/A280 ratio was determined with a NanoDrop; the A260/A280 ratios of all samples were in the approximate range of 1.9–2.1. The integrity of the RNA samples was assessed with an Agilent 2100 Bioanalyzer, and samples with no sign of degradation were selected for further analysis.

### Library construction and sequencing

A total of 9 samples, three per treatment, were randomly selected to constructed libraries. The first step in the workflow involved purifying the poly-A containing mRNA molecules
using poly-T oligo-attached magnetic beads. Following purification, the mRNA was fragmented into small pieces through divalent cations under elevated temperature. The cleaved RNA fragments were copied into first strand cDNA using reverse transcriptase and random primers. This was followed by second strand cDNA synthesis employing DNA Polymerase I and RNase H. A single 'A' base was then added to these cDNA fragments, and an adapter was subsequently ligated to the cDNA fragments. The products were then purified and enriched with PCR amplification. Then, the PCR yield was quantified by Qubit, and samples were pooled together to make a single-strand circular DNA (ssDNA circle), which was the final library. DNA nanoballs (DNBs) were generated with the circular ssDNA by rolling circle replication (RCR) to enlarge the fluorescent signals during the sequencing process. The DNBs were loaded into the patterned nanoarrays, and pair-end reads of 100 bp were read with the BGISEQ-500 platform for the following data analysis. For this step, the BGISEQ-500 platform combined the DNA nanoball-based nanoarrays and stepwise sequencing employing the combinational probe-anchor synthesis sequencing method. Raw reads containing a low quality, adaptor-polluted and a high content of unknown bases (N) were removed before downstream analyses. Reads cleaning and filtering were performed by BGI internal software.

## De novo assembly and functional annotation

Because published genome of safflower was not integrated, this project was done without a reference genome. Clean reads in each sample were assembled independently to obtain a reference sequence for subsequent analysis. De novo transcriptome assembly was completed by Trinity v2.0.6 (*Grabherr et al., 2011*) with clean reads, and then TGICL v2.0.6 (*Pertea et al., 2003*) was used on cluster transcripts to remove redundant transcripts and obtain Unigenes. The Unigenes were divided into two types: one type was a cluster in which the prefix was CL with the cluster ID behind it. In one cluster, there are several Unigenes which similarity between them is more than 70%. Another type was a singleton in which the prefix was Unigene. The N50, N70 and N90 length were used to determine the assembly continuity. The higher the length of N50, N70 and N90, the better the assembling continuity was. The raw data were uploaded to the NCBI and Figshare.

Unigene sequences were aligned to the following databases with BLAST (v2.2.23): NT (https://www.ncbi.nlm.nih.gov/nucleotide), NR (https://www.ncbi.nlm.nih.gov/refseq/about/nonredundantproteins/), COG (http://www.ncbi.nlm.nih.gov/COG), KEGG (http://www.genome.jp/kegg) and SwissProt (http://www.ebi.ac.uk/swissprot/). Based on the NR annotation, GO functional annotation (http://geneontology.org) was obtained through the Blast2GO program (v2.5.0). InterPro annotations (http://www.ebi.ac.uk/interpro) was obtained by InterProScan 5.

## Analysis of the differentially expressed genes

Clean reads were mapped to the Unigenes by Bowtie2 v2.2.5 (*Langmead & Salzberg, 2012*) and then the gene expression levels were calculated through RSEM v1.2.12 (*Li & Dewey, 2011*). Differential expression analysis was performed by DEseq2 as described by

*Love, Huber & Anders (2014)*. DEseq2 is a package implemented in R based on the negative binomial distribution. A fold change ≥2.00 and an adjusted *P* value ≤ 0.05 were set as the thresholds to establish significantly differential expression. Classification and functional enrichment analysis including Gene Ontology (GO) and KEGG were performed to identify which GO terms or metabolic pathways were significantly enriched in the Differentially expressed genes (DEGs).

### Real-time PCR expression analysis of DEGs

Because of the pharmacological effects of safflower flavonoid, we focused on DEGs related to flavonoid biosynthesis in this study. Real-time PCR (RT-PCR) analysis was employed to confirm the results of the RNA sequencing, and expressions of selected DEGs were measured in three different treatments. Each sample replicated 3 times. Specific primers were designed by Primer Premier 5 software (Table S1). Gene expression under different conditions was measured with the CFX96™ Real-time System (Bio-Rad, Hercules, CA, USA) using SYBR Premix Ex Taq II (TaKaRa, Japan). The *25S* rRNA gene obtained from *Carthamus tinctorius* L. was used as the reference gene to identify differences in each cDNA template. The RT-PCR cycling conditions were as follows: 95 °C for 3 min, followed by 40 cycles of 95 °C for 10 s and 61 °C for 30 s. The $2^{-\Delta\Delta Ct}$ method (*Livak & Schmittgen, 2001*) was employed to analyze relative gene expressions. ANOVA with a post-hoc Tukey-Test in SPSS (version 20) was used for data analysis. Three *HCT* genes, two *FLS* genes and two *ANS* genes related to flavonoid biosynthesis, namely, *CtHCT1*, *CtHCT2*, *CtHCT3*, *CtFLS1*, *CtFLS2*, *CtANS1* and *CtANS2*, were successfully amplified and chosen to certify the results.

### Flavonoid quantification

Total flavonoid content was determined by the colorimetric method as previously described (*Yu et al., 2010*) with some modifications. Powdered flower tissue (0.2 g) was dissolved in 10 ml of 50% (v/v) methanol and subjected to ultrasonic extraction for 90 min. Scanning in the wavelength range of 200–600 nm, the optimum absorption peak wavelength was 255 nm. Therefore, the absorbance of the solution was measured at 255 nm by SpectraMax iD3 (Molecular, USA). Each sample replicated 3 times. ANOVA with a post-hoc Tukey-Test in SPSS was used for data analysis.

To determine the HSYA content, 0.4 g of powdered flower tissue was dissolved in 50 ml of 25% (v/v) methanol and then subjected to ultrasonic extraction for 40 min. The samples were then diluted with methanol, filtered through a 0.45 μm filter membrane and analyzed by high performance liquid chromatography (HPLC) (Agilent 1200, Santa Clara, CA, USA). The HSYA content was analyzed with an Agilent C18 chromatographic column (4.6 mm × 250 mm, 5 μm) in conjunction with a mobile phase that consisted of methanol, acetonitrile and 0.7% (v/v) phosphoric acid (26:2:72) for elution. The flow rate was 0.8 ml min$^{-1}$, the injection volume was 10 μl and the UV detector was set such that λ = 407 nm. To determine the kaempferol content, 1.0 g of powdered flower tissue was dissolved in 25 ml of methanol and then subjected to heat reflux extraction at 95 °C for 30 min followed by hydrochloric acid hydrolysis for 30 min.

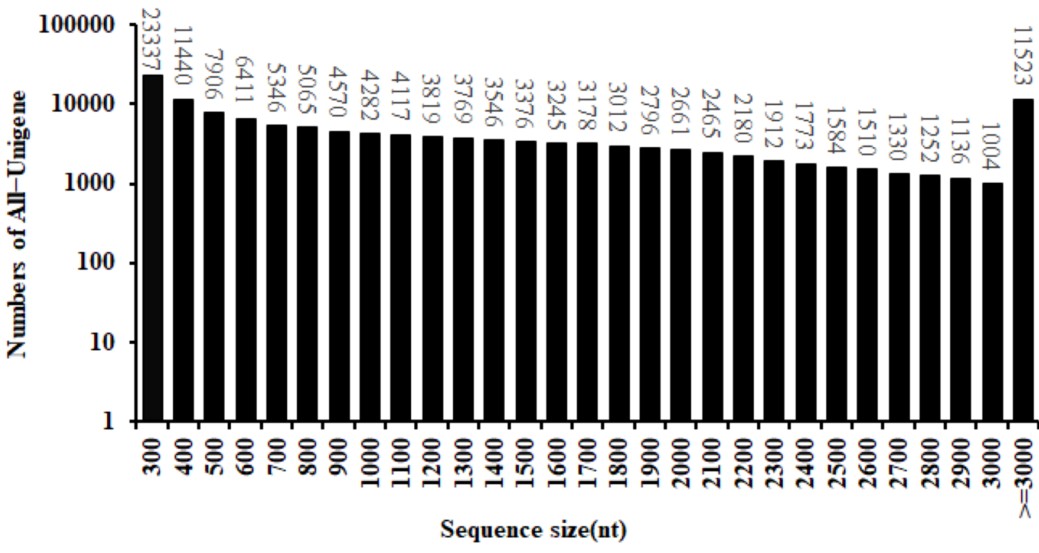

**Figure 2 Length distribution of All-Unigene in safflower transcriptome.**

The sample solutions were diluted with methanol, filtered through a 0.45 μm filter membrane and analyzed by HPLC. The kaempferol content was analyzed with an Agilent C18 chromatographic column in conjunction with a mobile phase consisting of methanol and 0.4% (v/v) phosphoric acid for elution. The flow rate was 1.0 ml min$^{-1}$, the injection volume was 20 μl and the UV detector was set such that λ = 360 nm and 254 nm. Each sample replicated 3 times. ANOVA with a post-hoc Tukey-Test in SPSS was used for data analysis.

# RESULTS

## Sequencing and functional annotation

By the use of the BGISEQ-500 high-throughput sequencing platform, a total of 99.16 Gb of clean data were obtained. The Q30 fraction of each sample was greater than or equal to 88.41%. Clean reads quality metrics were shown in Table S2. These results indicated that the quality of the sequencing data was sufficient to continue with the following analysis. 129,544 Unigenes were generated with a total length, mean length, N50 and GC content of 170,757,704 bp, 1,318 bp, 2,090 bp and 41.59%, respectively. The distribution of the Unigenes lengths was shown in Fig. 2. The Unigenes quality metrics were shown in Table 1.

Finally, a total of 78,179 Unigenes were annotated according to seven functional databases. Among all annotated Unigenes, 47,750 were annotated via the SwissProt database, 53,683 were annotated via the KEGG database, and 23,382 were annotated by the GO database (Table 2).

## Analysis of DEGs under different light intensities

Based on their gene expression levels, DEGs between groups were identified using the methods described above. A total of 1,938 DEGs were identified among all Unigenes.

**Table 1  Quality metrics of Unigenes.**

| Sample | Total number | Total length | Mean length | N50 | N70 | N90 | GC (%) |
|---|---|---|---|---|---|---|---|
| HL_1 | 47,956 | 48,017,129 | 1,001 | 1,597 | 1,038 | 422 | 43.49 |
| HL_2 | 42,932 | 40,607,825 | 945 | 1,520 | 961 | 388 | 43.43 |
| HL_3 | 42,861 | 40,604,125 | 947 | 1,514 | 959 | 391 | 43.6 |
| LL_1 | 72,845 | 71,641,113 | 983 | 1,631 | 1,007 | 391 | 41.59 |
| LL_2 | 59,271 | 56,435,088 | 952 | 1,589 | 987 | 376 | 42.65 |
| LL_3 | 61,471 | 58,975,317 | 959 | 1,594 | 998 | 380 | 42.47 |
| ML_1 | 51,245 | 51,521,358 | 1,005 | 1,626 | 1,054 | 417 | 43.03 |
| ML_2 | 48,597 | 49,086,125 | 1,010 | 1,619 | 1,059 | 421 | 43.33 |
| ML_3 | 60,239 | 53,701,642 | 891 | 1,510 | 896 | 344 | 42.54 |

Note:
N50, the N50 length is used to determine the assembly continuity, the higher the better; N70, similar to N50; N90, similar to N50; GC (%), the percentage of G and C bases in all Unigenes.

**Table 2  Unigene annotation and statistics for safflower de novo transcriptome.**

| Categories | Number | Frequency (%) |
|---|---|---|
| Total | 129,544 | 100 |
| NR | 70,977 | 54.79 |
| NT | 45,716 | 35.29 |
| SwissProt | 47,750 | 36.86 |
| KEGG | 53,683 | 41.44 |
| KOG | 58,385 | 45.07 |
| Interpro | 61,507 | 47.48 |
| GO | 23,382 | 18.05 |
| All annotated Unigenes | 78,179 | 60.35 |

Comparison of the DEGs in the three different light intensity conditions revealed 1,249, 242 and 698 DEGs when the following conditions were compared: HL vs LL, HL vs ML and ML vs LL, respectively. The distributions of the upregulated and downregulated Unigenes are shown in Fig. 3.

According to GO functional analysis and significant enrichment analysis of the DEGs, three comparisons showed a similar functional enrichment (Fig. 4). The "biological process" category was the most enriched, followed by the "cellular component" category. In the biological process category, the "metabolic processes", "cellular processes" and "single-organism processes" terms were significantly enriched. In the cellular component category, "cell", "cell part" and "membrane" were the three largest categories. In the molecular function category, the two largest categories were "binding" and "catalytic activity".

According to KEGG functional enrichment analysis, the "Metabolism" pathway was the most enriched pathway in all three comparisons. A total of 110 relevant metabolic pathways were identified by KEGG pathway analysis (Fig. 4).

To further investigate the Unigenes involved in flavonoid biosynthesis under different light conditions, 13 DEGs related to flavonoid biosynthesis were screened and identified through GO and KEGG (Table S3). Their FPKM (Fragments Per Kilobase of transcript per
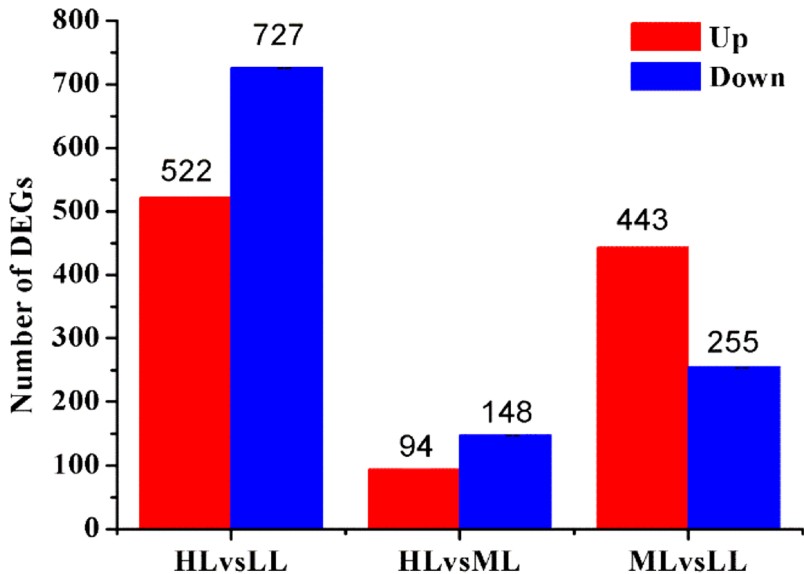

**Figure 3 Statistic of differently expressed genes obtained by DEseq2.** Red color represents up regulated DEGs. Blue color represents down regulated DEGs. HL, High light; ML, Middle light; LL, Low light.                                                               

Million fragments mapped) were showed in Fig. 5 and sequence data were listed in Supplemental Files.

## RT-PCR analysis of DEGs involved in flavonoid biosynthesis

A total of 7 of 13 DEGs related to flavonoid biosynthesis were successfully confirmed by RT-PCR. *CtHCT1* and *CtHCT2* were significantly downregulated in the LL condition compared to their expressions in the other groups. However, *CtHCT3* was downregulated in ML and upregulated in LL. Regarding the *FLS* and *ANS* genes, *CtFLS1* and *CtANS1* were upregulated in LL, but *CtFLS2* and *CtANS2* were downregulated in LL compared to their expressions in other treatment groups (Fig. 6). These results indicated that a decrease in light intensity had a remarkable impact on the expressions of flavonoid biosynthesis related genes in safflower flowers, and the decrease in light intensity to different degrees resulted in different effects on the expressions of homologous genes.

## Analysis of flavonoid content under different light intensities

The total flavonoid content reached a maximum in the ML condition, followed by that in the HL condition, and the lowest flavonoid content was measured in the LL condition. However, the differences of total flavonoid contents among three conditions were not significant. The HSYA content showed little change between the ML and LL conditions but was significant decreased in the HL condition. However, kaempferol showed a different accumulation pattern in three conditions compared with HSYA; it was the highest in the ML condition and was distinctly lower in both the HL and LL conditions (Fig. 7). These results indicated that a reduction in the light intensity within a suitable range was advantageous for flavonoid accumulation in safflower. In addition, the kaempferol

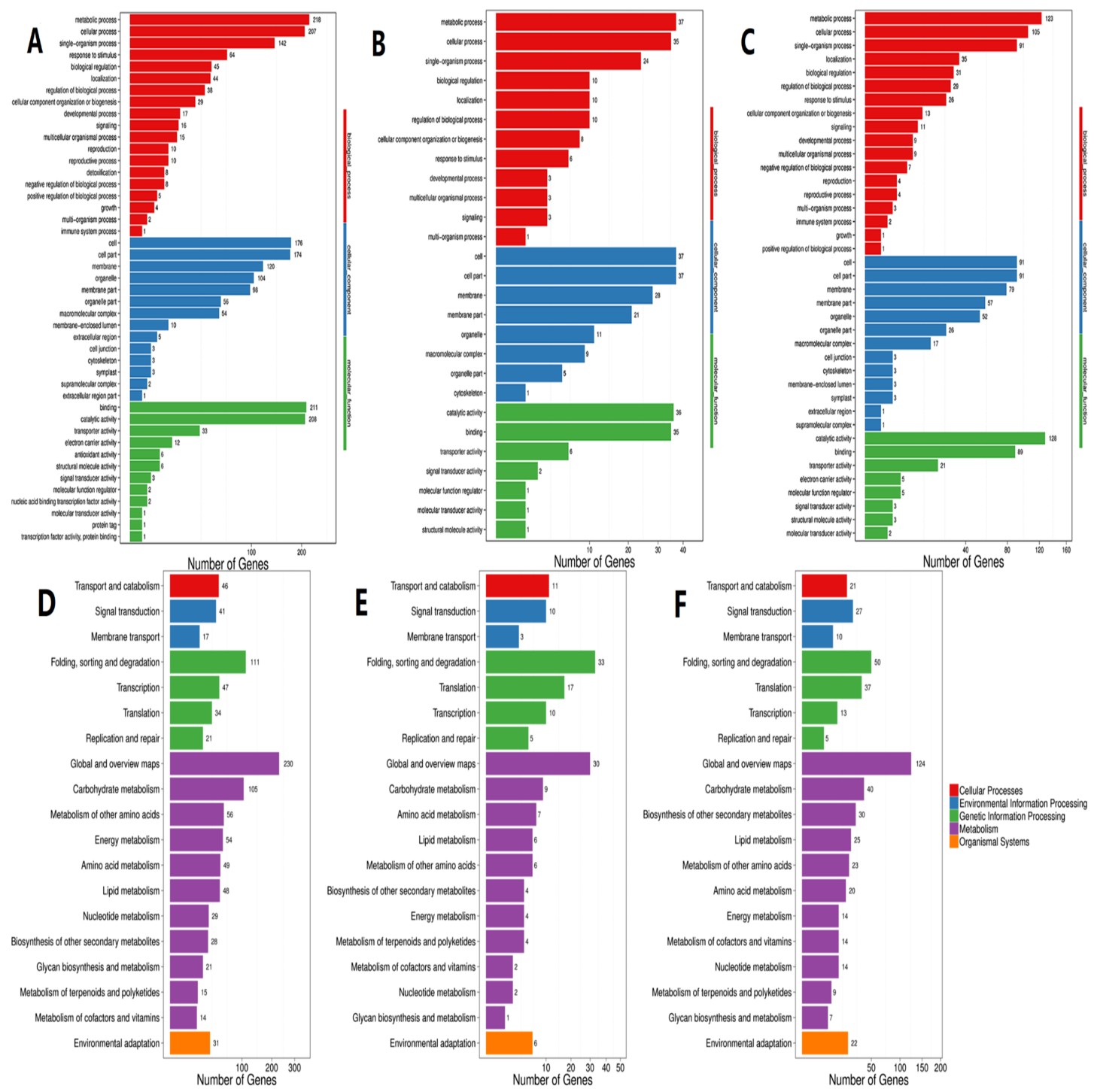

**Figure 4 GO and KEGG pathway classifications of DEGs.** *X* axis represents number of DEG. *Y* axis represents GO term in (A)–(C): (A) comparison of HL vs. LL, (B) comparison of HL vs. ML, (C) comparison of ML vs. LL; *Y* axis represents functional classification of KEGG in (D)–(F): (D) comparison of HL vs. LL, (E) comparison of HL vs ML, (F) comparison of ML vs. LL.

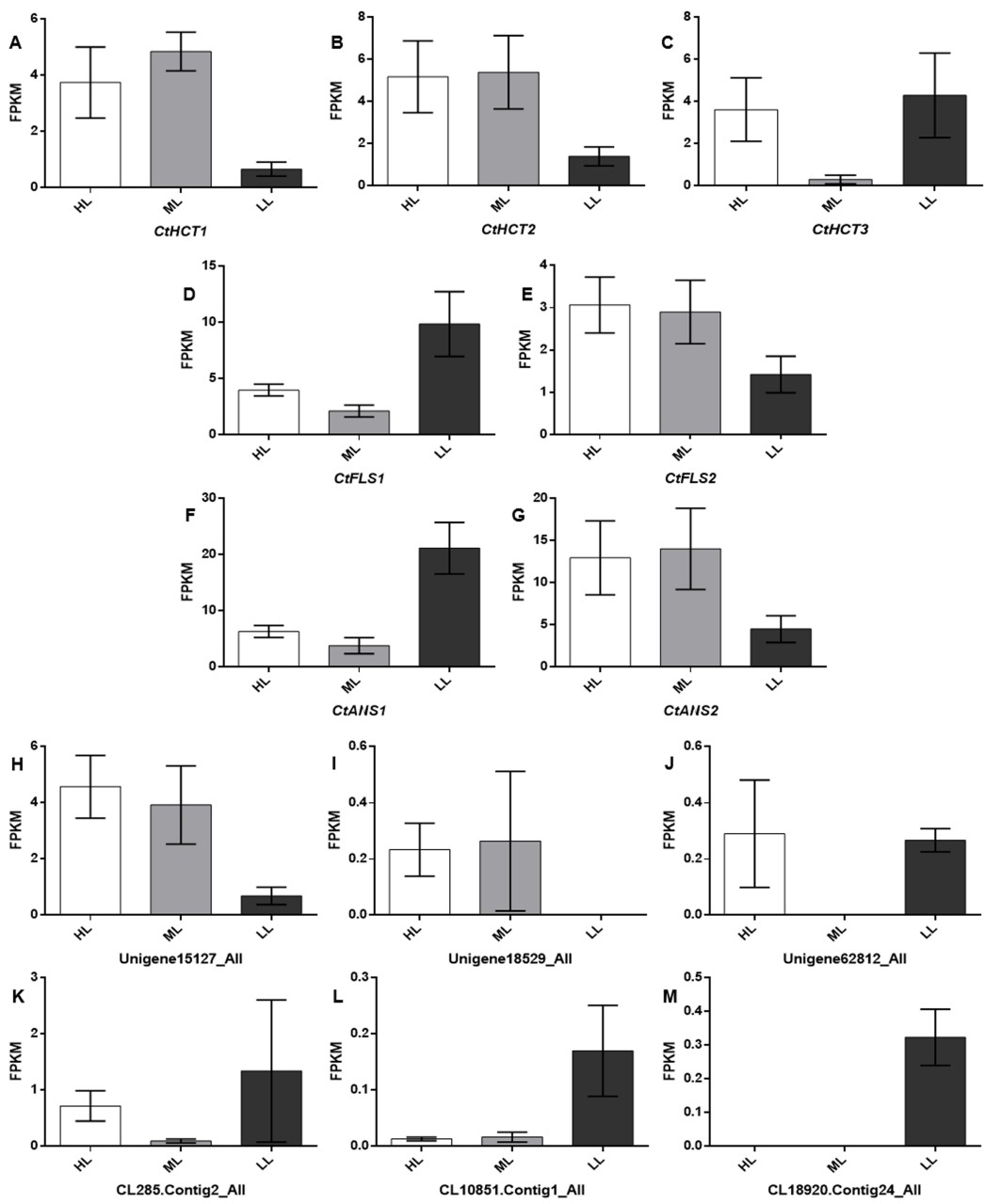

**Figure 5 Expression levels of the DEGs related to flavonoid biosynthesis.** A total of 13 selected differential expressed genes related to flavonoid biosynthesis from the transcriptome and average FKPM values for biological replicates are shown. (A–M) *CtHCT1*, *CtHCT2*, *CtHCT3*, *CtFLS1*, *CtFLS2*, *CtANS1*, *CtANS2*, Unigene15127_All, Unigene18529_All, Unigene62812_All, CL285.Contig2_All, CL10851. Contig1_All and CL18920.Contig24_All, respectively. HL, ML and LL presented high light intensity, middle light intensity and low light intensity respectively.

concentration showed a significant negative relationship with *CtHCT3* expression ($P < 0.01$), but no significant relationships with other DEGs. The authors suggest that flavonoid biosynthesis was affected by the differential expression of *HCT* genes, especially *CtHCT3*, in safflower flowers grown under different light intensities.

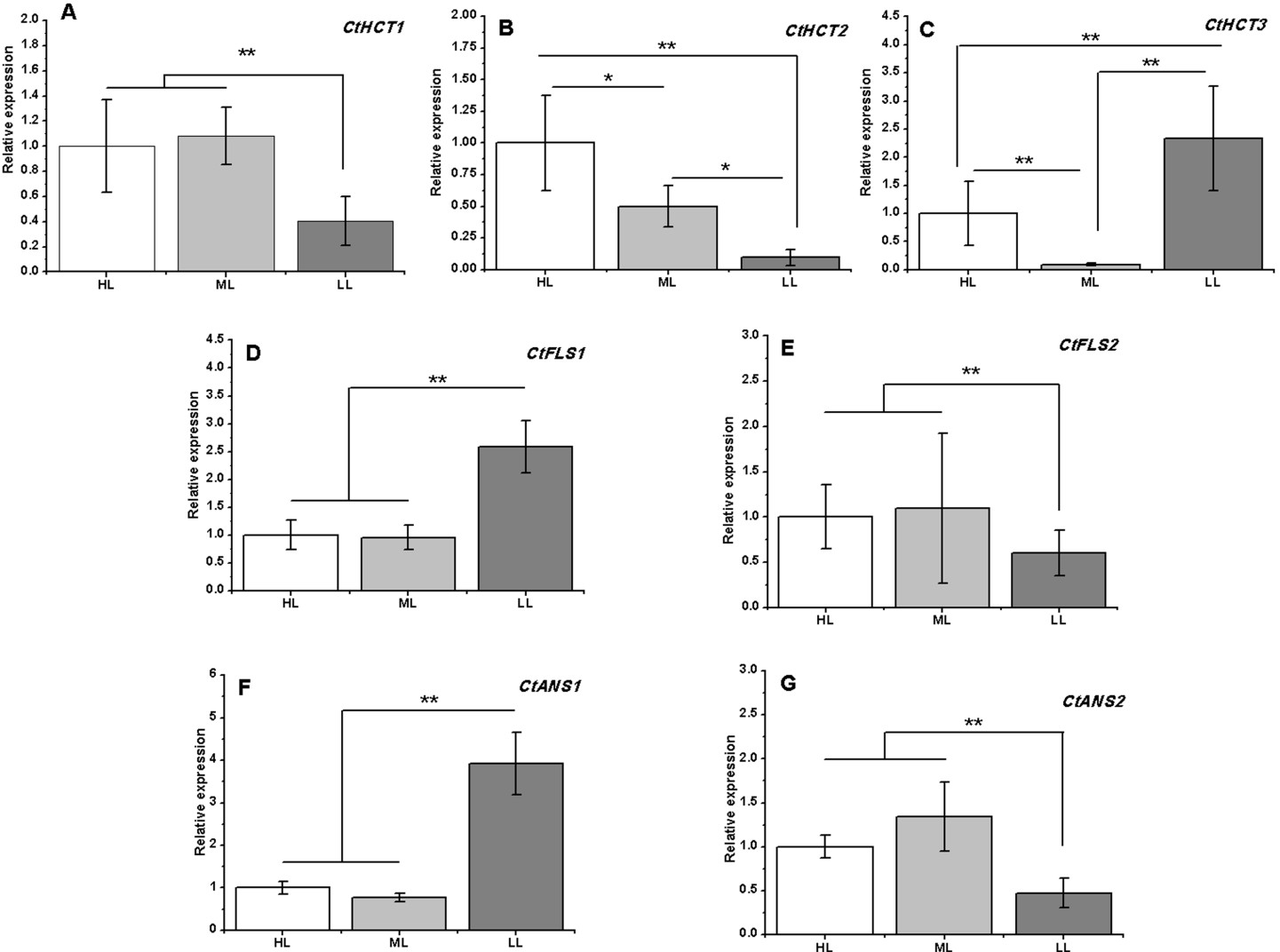

**Figure 6 Expression analysis by RT-PCR of seven flavonoid related genes under different light intensities.** (A)–(G) *CtHCT1*, *CtHCT2*, *CtHCT3*, *CtFLS1*, *CtFLS2*, *CtANS1* and *CtANS2*, respectively. Results of ANOVA with a post-hoc Tukey-Test are shown with asterisks (*$P < 0.05$; **$P < 0.01$).

## DISCUSSION

The molecular mechanisms of flavonoid biosynthesis in safflower have not yet been defined. *Huang et al. (2012)* used Illumina-based de novo transcriptome sequencing to discover all known genes and primary metabolic pathways in this transcriptome. A total of 156 Unigenes that encode enzymes involved in flavonoid synthesis were identified based on the KEGG annotation. *Liu et al. (2015)* used 454 pyrosequencing to investigate genes related to the biosynthesis of safflor yellow, and 22 Unigenes involved in flavonoid biosynthesis were identified. In our previous research (*Chen et al., 2018*), by three-generation sequencing (PacBio RS II platform), 44 unique isoforms encoding enzymes involved in flavonoid biosynthesis were screened. However, 104 Unigenes were related with flavonoid biosynthesis in this experiment and 13 DEGs related with flavonoids

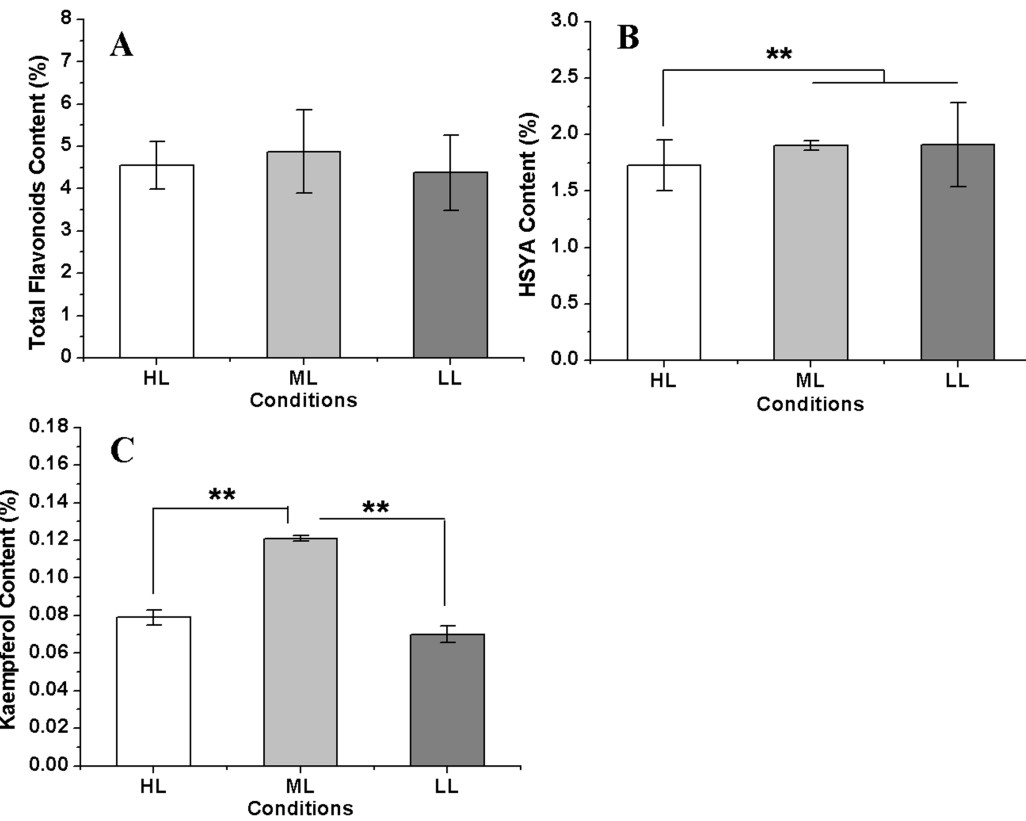

**Figure 7 The flavonoids concentration in safflower flowers under different light intensities.** (A)–(C) Contents of total flavonoids, hydroxysafflor yellow A (HSYA) and kaempferol, respectively. Results of ANOVA with a post-hoc Tukey-Test are shown with asterisks (**$P < 0.01$).

biosynthesis were not found in all above studies. Due to different depths of sequencing, the deeper depth of the sequencing is, the more functional genes that will be annotated. For instance, 99.16 Gb of data were obtained in this experiment, but only 4.69 Gb of clean reads was obtained by *Huang et al. (2012)* and 10.43 Gb of clean data was obtained by three-generation sequencing. On the other hand, it was the first time that the relationship between light intensity and safflower flavonoid biosynthesis had been studied. Accordingly, seven DEGs confirmed in this research had not been matched with previous researches.

In some species, increased light intensity can promote flavonoid accumulation (*Deng et al., 2012*; *Wang et al., 2018*). However, some species can increase the content of flavonoid under low light intensity to some extent (*Lu et al., 2017*; *Zhu, 2010*). In this experiment, although safflower growth was better in high light intensity, flavonoid content reached the highest level under ML condition.

In *Arabidopsis thaliana*, HCT repression had a major impact on phenylpropanoid metabolism and resulted in the redirection of metabolic flux towards the flavonoid pathway (*Besseau et al., 2007*). In our previous study (*Chen et al., 2018*), 14 *HCT* genes were identified from safflower via full-length sequencing. Three of 14 *HCT* genes

responded to MeJA treatment, and two of them were significantly regulated by MeJA. Accordingly, the contents of many flavonoids were significantly stimulated under MeJA treatment. It was indicated that *HCT* genes were closely related to flavonoid biosynthesis in safflower.

In this experiment, *CtHCT1* and *CtHCT2* were downregulated under LL conditions, but *CtHCT3* was upregulated under LL conditions. Accordingly, the HSYA content was the highest in the LL condition, but the total flavonoid and kaempferol concentrations were the lowest in the LL condition. In contrast, *CtHCT3* was downregulated in the ML condition. Accordingly, the contents of total flavonoid, HSYA and kaempferol increased in the ML condition. In addition, the kaempferol content showed a significant negative relationship with *CtHCT3* expression. Therefore, we believe that the downregulation of *CtHCT3* may have a positive effect on flavonoid accumulation. On the other hand, phylogenetic relationship among three *HCT* genes showed that *CtHCT1* and *CtHCT2* had closer genetic relationship with each other than *CtHCT3* (Fig. S1). Thus, we suggested that three *HCT* genes in safflower had different subfunction: *CtHCT1* and *CtHCT2* may had protective function under high light intensity, and *CtHCT3* might be closely related to flavonoid biosynthesis.

However, the different trends observed for HSYA and kaempferol accumulation might be attributed to their different synthesis pathways and effects from other genes such as *FLS* and *ANS*. The expression levels of three *HCT*, two *FLS* and two *ANS* genes in safflower flowers grown under different light intensities were analysed, but which homologous gene played a major role in flavonoid biosynthesis requires further study.

Through the above experiments, we suggested that middle light intensity could promotes flavonoids accumulation in safflower. We hypothesized that a decrease in light intensity led to the downregulation of *HCT* genes in safflower, then the downregulation of *HCT* genes might result in the repression of lignin synthesis, increased substrate catalysis by CHS and promoting flavonoid accumulation. It was also found that the growth of safflower plants was inhibited when the light intensity was reduced, which might have affected *HCT* expression and the repression of lignin synthesis. Therefore, the authors believe that safflower grew vigorously under high light conditions, but a reduction of the light intensity within a suitable range promoted flavonoid biosynthesis in safflower flowers.

## CONCLUSIONS

This is the first transcriptome analysis of *Carthamus tinctorius* L. exposed to different light conditions. We obtained a total of 99.16 Gb of clean reads. A total of 129,544 Unigenes were identified, 78,179 (60.3%) of which were functionally annotated. A considerable number of high-quality Unigenes were identified that will be useful for future studies. A total of 13 novel genes related to flavonoid biosynthesis were identified based on the functional annotation of differentially expressed genes, and the differential expressions of seven of them under different light intensities were analyzed by RT-PCR. The concentrations of total flavonoid and two active compounds were measured in safflower flowers grown under three different light conditions. We confirmed that the expressions of *HCT* genes played an important role in flavonoid biosynthesis in safflower

flowers grown under different light intensities. Our study lays a foundation for further research on the effects of light on flavonoid biosynthesis in safflower flowers.

## ACKNOWLEDGEMENTS

Many thanks go to the participating families and the entire study team.

### Funding

This work was supported by the National Natural Science Foundation of China (Nos. 81573544, 81803669 and 81703654), the Department of Science and Technology of Sichuan Province (No. 2014SZ0156), and Chengdu University of Traditional Chinese Medicine (Nos. ZRYY1610 and ZRQN1647). The funders had no role in study design, data collection and analysis, decision to publish, or preparation of the manuscript.

### Grant Disclosures

The following grant information was disclosed by the authors:
National Natural Science Foundation of China: 81573544, 81803669 and 81703654.
Department of Science and Technology of Sichuan Province: 2014SZ0156.
Chengdu University of Traditional Chinese Medicine: ZRYY1610 and ZRQN1647.

### Competing Interests

The authors declare that they have no competing interests.

### Author Contributions

- Chaoxiang Ren conceived and designed the experiments, performed the experiments, analyzed the data, prepared figures and/or tables, authored or reviewed drafts of the paper, and approved the final draft.
- Jie Wang performed the experiments, prepared figures and/or tables, and approved the final draft.
- Bin Xian performed the experiments, prepared figures and/or tables, and approved the final draft.
- Xiaohui Tang performed the experiments, prepared figures and/or tables, and approved the final draft.
- Xuyun Liu performed the experiments, authored or reviewed drafts of the paper, and approved the final draft.
- Xueli Hu performed the experiments, authored or reviewed drafts of the paper, and approved the final draft.
- Zunhong Hu performed the experiments, authored or reviewed drafts of the paper, and approved the final draft.
- Yiyun Wu performed the experiments, prepared figures and/or tables, and approved the final draft.
- Cuiping Chen analyzed the data, authored or reviewed drafts of the paper, and approved the final draft.

- Qinghua Wu analyzed the data, authored or reviewed drafts of the paper, and approved the final draft.
- Jiang Chen conceived and designed the experiments, authored or reviewed drafts of the paper, and approved the final draft.
- Jin Pei conceived and designed the experiments, authored or reviewed drafts of the paper, and approved the final draft.

## Data Availability

The transcriptome sequencing raw data are available at NCBI and Figshare: SRR7943406, SRR7943405, SRR7943404, SRR7943403, SRR7943402, SRR7943401, SRR7943400, SRR7943399, SRR7943407.

Ren, Chaoxiang (2019): HS3_1_1.fq.gz. figshare. Dataset. DOI 10.6084/m9.figshare.8281379.v1.

Ren, Chaoxiang (2019): HS3_1_2.fq.gz. figshare. Dataset. DOI 10.6084/m9.figshare.8281382.v1.

Ren, Chaoxiang (2019): HS3_2_1.fq.gz. figshare. Dataset. DOI 10.6084/m9.figshare.8281103.v1.

Ren, Chaoxiang (2019): HS3_2_2.fq.gz. figshare. Dataset. DOI 10.6084/m9.figshare.8281109.v1.

Wang, Jie (2019): HS3_3_1.fq.gz. figshare. Dataset. DOI 10.6084/m9.figshare.8282114.v1.

Wang, Jie (2019): HS3_3_2.fq.gz. figshare. Dataset. DOI 10.6084/m9.figshare.8282117.v1.

Wang, Jie (2019): LS3_1_1.fq.gz. figshare. Dataset. DOI 10.6084/m9.figshare.8282120.v1.

Wang, Jie (2019): LS3_1_2.fq.gz. figshare. Dataset. DOI 10.6084/m9.figshare.8282123.v1.

Ren, Chaoxiang (2019): MS3_1_1.fq.gz. figshare. Dataset. DOI 10.6084/m9.figshare.8287691.v2.

Ren, Chaoxiang (2019): MS3_1_2.fq.gz. figshare. Dataset. DOI 10.6084/m9.figshare.8287694.v1.

Ren, Chaoxiang (2019): MS3_2_1.fq.gz. figshare. Dataset. DOI 10.6084/m9.figshare.8287697.v1.

Ren, Chaoxiang (2019): MS3_2_2.fq.gz. figshare. Dataset. DOI 10.6084/m9.figshare.8287700.v1.

Xian, Bin (2019): MS3_3_1.fq.gz. figshare. Dataset. DOI 10.6084/m9.figshare.8292788.v1.

Xian, Bin (2019): MS3_3_2.fq.gz. figshare. Dataset. DOI 10.6084/m9.figshare.8292791.v1.

Ren, Chaoxiang (2019): LS3_2_1.fq.gz. figshare. Dataset. DOI 10.6084/m9.figshare.8283188.v1.
## Supplemental Information

Supplemental information for this article can be found online at http://dx.doi.org/10.7717/peerj.8671#supplemental-information.

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
