# Peer review of "Transcriptome analysis of flavonoid biosynthesis in safflower flowers grown under different light intensities"

_PeerJ, doi:10.7717/peerj.8671_

## Round 0.1 · original submission · Major Revisions

Dear authors

As you can see, your ms has been read carefully by 2 revievers. They recommend a thorough revision. Make sure that you change your ms accordingly as these reviewers will check your revision

Kind regards
Michael Wink
Editor

Reviewer 1 ·

Basic reporting

The paper “Transcriptome analysis of flavonoid accumulation in safflower flowers grown under different light intensities” is well written, but still needs some work to make it clearer. For instance, in the introduction a phrase linking the first and the second paragraphs is missing.
Some paragraphs later in the results part can be cut out.

The background is well researched.

The raw data is shared, the link to figshare is missing.

It is self-contained, but the hypothesis needs to be stated more clearly.

Experimental design

As mentioned above, the hypothesis and research question needs to be stated more clearly.

The main hypothesis, i.e. the relation of the light intensity UV protection and flavonoid content is unfortunately not clearly highlighted and explained in the paper. For instance, in the “Background”, following phrase reads: “Flavonoids are the main active ingredients in safflower and have many pharmacological effects.” It might be worth adding something like: “and play a role in UV light protection”. Multiple papers have been published on the subject. The missing hypothesis is more clear in p6. Lines 41-45, were the experimental setup to study relationship between light and flavonoid production is not clearly described.

Here I state some of the missing parts, that can provide more detail:

p.6, l. 51. “plant materials used in this study were safflower … obtained” something is missing here: seeds? Leafs? Complete plants?

p.6, l. 59. “Safflower plants with the same growth status” Please indicate the age/time here, for instance 5 weeks old… also, how long were the plants grown in the different light treatments? Please indicate!

p.6, l.63. Materials and Methods: please cite (or describe) here which RNA extraction protocol you used.

Validity of the findings

If the authors address most of my concerns then the paper will be a novel and robust paper, with an original research question and well supporting results.

Additional comments

The paper “Transcriptome analysis of flavonoid accumulation in safflower flowers grown under different light intensities” is well written. It is an exploration of the flavonoid biosynthesis genes under different light regimes, which is a complement to previous more descriptive transcript analyses. The role of flavonoids in plant UV protection and a probable link and/or tradeoff with secondary metabolite production is a question that is still not been extensively explored.
The main hypothesis, i.e. the relation of the light intensity UV protection and flavonoid content is unfortunately not clearly highlighted and explained in the paper. For instance, in the “Background”, following phrase reads: “Flavonoids are the main active ingredients in safflower and have many pharmacological effects.” It might be worth adding something like: “and play a role in UV light protection”. Multiple papers have been published on the subject. The missing hypothesis is more clear in p6. Lines 41-45, were the experimental setup to study relationship between light and flavonoid production is not clearly described.

Detailed important comments and possible corrections:
p.5, l.3. “Safflower is a strongly domesticated species” what does strongly mean here? i.e. underwent a strong bottleneck? From this phrase it’s not clear what the authors mean.

p.5, l.16-17. A transition phrase between paragraph 1 (l.16) and paragraph 2 is not existing. Please spell out a logical transition.

p.6, l. 51. “plant materials used in this study were safflower … obtained” something is missing here: seeds? Leafs? Complete plants?

p.6, l. 59. “Safflower plants with the same growth status” Please indicate the age/time here, for instance 5 weeks old… also, how long were the plants grown in the different light treatments? Please indicate!

p.6, l.63. Materials and Methods: please cite (or describe) here which RNA extraction protocol you used.

p.7, l.72. complete raw reads were eliminated? In most pipelines, only ends - both containing adaptors and/or low quality are trimmed from the sequences.

p.7, l.88. TGICL is quite an old tool therefore I suggest that they use a newer one, like BUSCO. Also, did they re-map the annotated data to the denovo transcriptome? this should be done to have an estimation of transcript representation of the transcriptome.

p.8, l.118. please indicate what type of spectrophotometer.

p.8, l.135. are the 78.8 Gb total? per sample? I guess total from Table 1. Please spell out.

p.9, paragraph functional annotation and classification. I would only mention the best annotation and swissprot and GO terms and include this in a phrase in the former paragraph. Not in a complete paragraph, IMO.


p.10, line 208-209. …two of them were significantly regulated by MeJA…. Please indicate which of the HMT genes were regulated in the text.

To include in the discussion:
How were the light treatments decided upon? The authors say: “The HL (high light) group
was grown under 40,000 Lux. The second group, ML (middle light), was grown under 20,000 Lux. The third group, LL (low light), was grown under 10,000 Lux.” Are this light conditions reflecting conditions at natural settings (which can vary between 2,000 Lux in the shadow and 120,000 Lux https://en.m.wikipedia.org/wiki/Daylight) ? Or rather reflecting illumination in greenhouses?


A discussion on the possibility of subfunctionalization of the HCT genes is missing, as HCT1 and HCT2 are upregulated in the HL condition (protective function), whereas HCT3 seems to be upregulated in the low light condition, a classical example of subfunctionalization.
It would be very informative to see the phylogenetic relationship among the three HTC genes, that seem to have undergone a subfunctionalization, as HCT1 and HCT2 are upregulated in the HL condition (protective function), whereas HCT3 seems to be upregulated in the low light condition, which might indicate a functional switch.

Suggestions for Figures and Tables:
Table 1: quality data should go into a supplementary table, so please move this Figure to the supplementals. Instead a Figure including the analyzed genes (HCT, FLS, ANS...) from the flavonin pathway in safflower would be useful.
Figure 3 and Figure 4 should be put together into one Figure.

Annotated reviews are not available for download in order to protect the identity of reviewers who chose to remain anonymous.

Reviewer 2 ·

Basic reporting

-Title it is not very appropriate. With a transcriptome analysis it is impossible to “measure” flavonoid accumulation, so “Transcriptome analysis of flavonoid accumulation” it is not correct.

-For a better understanding of a broad audience, introduction needs more background about the structure of the phenylpropanoid pathway and specially of the anthocyanin biosynthetic pathway (hereafter ABP) and how flavonoids compounds are synthesized. Maybe a figure it would be interesting.

-Information in Lines 11-16 are not very interesting for this study.

-Lines 17-24 describe extensively a previous work of the authors, however there are too much details of that paper that are unnecessary in the context of the present study. It can be resumed in this sentence: “light factors played an important role in controlling the distribution boundary of safflower (Wu, 2017).”

-Lines 25-35 focus on the pharmacological effects of flavonoids of safflower however this is not the aim of this study and it can be also resumed. In addition to the pharmacological effects it would be more interesting previous knowledge of the ABP in Carthamus: which enzymes are involved, which compounds are known, how are they synthesized, etc.

-It would be also important to include previous knowledge of the effect of light intensity and other environmental factors in flavonoid production or accumulation. Maybe there is no information for Carthamus, but there is in many other species.

-On the other hand, why do the authors use transcriptomics to answer their questions? Why is this methodology appropriate for their study? Maybe some phrases about this topic are needed.

-Figures and tables are of high quality and well labelled in general. However the description of some of them is not very accurate.

-Table S2 should be included in the main text (with some formatting), this information is one of the most important results of the paper. Or maybe a bars graphic with the expression values of those genes with differential expression.

-In Table 2, the full explanation of the N50 parameter it should be in M&M, in the table description only a short reminder.
- In Figures 3 & 4 it would be nice a larger font size for the y axis if possible. It would be helpful if the comparison (e.g. HLvsML) could be added above or below the figures.
- Figure 5: Since the actual fill of the bars does not give any information, it would be more informative to colour the bars by treatment (HL,ML and LL). Explain what the asterisks mean and which test was performed in the figure legend. In the figure legend, results should not be described.
- Figure 6: Since B and C are results from HPLC I suggest to put them together instead of A and B. On the other hand, instead of describing the results in the figure legend, describe which method was used for each quantification. As for Figure 5, explain what asterisks mean.

-Line 88: do you mean redundant?

-Lines 139-140: From which assembly come this results? From the new reference created? (line 86). Clarify this.

-Lines 168-171: This phrases should be in M&M. Describe what HCT, FLS and ANS means.

-Lines 180-181: I suppose that some tests were performed to test for differences in total flavonoid content but differences were not significant. Add this information in text.

-Lines 193-200: It would be interesting to know which enzymes were identified. There were DEG analyses in those papers? If so, it would be very interesting to know the results to compare with your results.

-Lines 207-210: Is this related with light intensity somehow? Explain

-In the discussion: How this new results fit with previous knowledge? How light intensity affects the synthesis of flavonoid in other species? Are these findings similar to previous results found?

Experimental design

-Lines 42-43: “To explore the relationship between light intensity and flavonoid accumulation in safflower flowers, transcriptome analysis was performed in this study”. Like a previous comment on the title, with a transcriptome analysis, flavonoid accumulation cannot be measured. The transcriptome experiment is useful to obtain the genes sequences and allows the digital expression analyses, but not the quantification of the flavonoids. For the quantification of flavonoids HPLC, the spectrophotometer or even photographs (in the case of anthocyanins) make it possible.

-Lines 46-47: “The correlation between gene expression and the flavonoid content was analyzed”. How was this analyzed? To correlate the expression analysis with the flavonoid content, those genes used for the expression analysis should be the responsible of the synthesis of those flavonoids quantified. This is true for the FLS and kaempferol but not for the HSYA.

-Line 60 specifies that 3 biological replicates were used for each treatment (I assume it is for the transcriptome experiment). However it is unclear how many libraries were finally constructed. The reader has to wait to results to see in table 1 that there was nine samples, three per treatment. In addition I can’t find how many replicates were used for the expression analysis with the RT-PCR or for the flavonoid quantification.

-Lines 85-86: I am not familiar with knowledge in Carthamus, however a quick search in genbank showed me that there is a genome available. Why did the authors performed a de novo assembly instead of using the genome as reference? I’m sure there are good reasons, however I think it should be justified. Another question arises from this lines, how was this new reference built? With all the libraries mixed? Or each sample was assembled independently and then a step of clusterization was performed? Clarify this.

-In general a better description of the methods should be made: which parameters were used when using the different programs (the default?, or customized ones?), which version of the software ( e.g line 87, line 97), how many replicates, etc. Or if authors are following a protocol from another paper it should be cited.

-Line 86: which program was used for read cleaning and filtering? Which criteria was used?

-Lines 86-91: I’m not sure to understand this paragraph. If TGICL was used to remove clusters of redundant transcripts and obtain unigenes, why unigenes were divided into two types: clusters and singletons? Rephrase or clarify this.

-Line 98: Specify RSEM version, and cite the authors of the software. DESeq2 is a package implemented in R? If so, this should be mentioned.

-Line 107: How were this genes selected and why? This should be described here, not in Results. How many samples and replicates? Is CHS the closest known enzyme that leads to the synthesis of HSYA? If so, why it hasn’t been included in the expression analyses?

-Line 114: “Data was analysed using SPSS” not “data were generated using SPSS”. Which tests were performed to find differences between treatments?

-Line 118: absorbance was measured at 255nm for total flavonoid content. Is this absorbance commonly used in Carthamus? Flavonols and flavons use to be measured at 300 or 350nm, and anthocyanins at 520nm. I find 255nm a bit low, however if it is correct please cite some literature.

-Lines 164-166: A figure with the digital expression values of these genes is essential (in adittion to statistical tests) to compare the digital expression values with the RT-PCR results.

Validity of the findings

- Please clarify if there are replicates in the expression and flavonoid quantification experiments
-How are the digital expression values within treatments? Are they regular?
-Lines 116-132: How authors test for differences between treatments? Which tests were used? There were replicates? How many?
-Lines 224-226: To say that CHS increases its substrate catalysis, CHS should be included in the expression analysis as well as the product (any chalcone) of its action in the flavonoid quatification. Why it wasn’t included?

---

## Round 0.2 · Minor Revisions

Dear authors

As you can see, our reviewers still found several points which need your attention. Hope you can revise your ms accordingly

Regards
M. Wink
Editor

Reviewer 1 ·

Basic reporting

Many things are not clear and ambigous. See review.

References are missing to be able to better asses versions of software used, etc.

Article structure not very professional, Figure and Table captions are too brief to decipher the information, Quality of Figures is not good. Labeling in Figures too small to read.

Data are shared on NCBI and figshare. Figshare link is missing.

Yes, although the minimum was made and the importance of the transcriptome is not very clear.

Experimental design

As mentioned in the review, for qPCR the 2-ΔΔCT method (Livak & Schmittgen, 2001) might be inappropiate.
I think it’s a problem to use of 2 as the efficiency base, this assumes perfect binding of the primers to a base 2 amplification per cycle. Instead the formula is E-ΔΔCT, where the primer efficiency needs to be calculated as described in Medrano & Wong 2018.

See above.

Since light appears to be the essential factor in the experiment, information on the chamber type- lamps might be needed to replicate the experiments.

Validity of the findings

Corrections need to be made to increase the validity.

Additional comments

Dear authors,

Please find enclosed an extensive review of your re-submitted paper: Transcriptome analysis of flavonoid biosynthesis in safflower flowers grown under different light intensities. Most of these are formulated as questions, hopefully this helps.

Review:
Main text :

L. 18- 19. In this study, we aimed to explore the relationship between different greenhouse light intensities used for growth and

L. 13 : Please remove the strongly and replace by “ancient” or a more precise meaning, what do they mean ? « widely used » ? Strongly implies that there’s low genetic variability within the cultivar, which I think is not really the case.

L. 142 : Why did they not use the safflower genome at least to remap the transcriptome data to. Is there a logical explanation?

L. 156 : The information on the version of the different databases KEGG, SwissProt is missing !

L. 176 : The 2-ΔΔCT method (Livak & Schmittgen, 2001) was employed to analyze relative gene expressions.
I think it’s a problem to use of 2 as the efficiency base, this assumes perfect binding of the primers to a base 2 amplification per cycle. Instead the formula is E-ΔΔCT, where the efficiency needs to be calculated as described in Medrano & Wong 2018 (https://doi.org/10.2144/05391RV01)

L. 245, 287, 289 : Total flavonol contents are said to be higher, then to be lower and then to “peak”, this is not true, as total flavonoid contents do not display significant differences among the 3 conditions.

L. 291- 292 : « Therefore, we believe that the downregulation of HCT may have a positive effect on flavonoid accumulation.” Do they mean CtHCT3? Else the statement is not clear.

L. 302- 305. “Through the above experiments, we suggested that a decrease in light intensity led to the downregulation of HCT genes in safflower, then the downregulation of HCT genes resulted in the repression of lignin synthesis and increased substrate catalysis by CHS, promoting flavonoid accumulation.” But this is not shown in the experiments! ML promotes flavonoids and this is not what this statement implies.


Comments on Figures and Tables :
The captions are too brief. Many times more information is needed for readers that will only look at the Figures. For instance :

Table 2 » Unigene annotation and assembly statistics.” Can be completed with: “for safflower de novo transcriptome assembly using Trinity v. 2.2.3”
The same is valid for all other Figures.

Fig. 1: “Proposed a part of flavonoid biosynthesis pathway in safflower”, showing enzymes involved in kaempferol and HCYA biosynthesis analyzed in this work. HCT= Hydroxi….; CHS= chalcone synthase; CHI=…; FLS=…..
At least the enzymes discussed in the main text and leading to the compounds of interest should be highlighted.

Fig. 2 is uninformative

Fig. 3” Statistic of differently expressed genes” obtained by DESeq2 (version X ) in safflower transcriptome subject to different light regimes HL= High light, ML=… LL=.

Fig. 4 Descriptions in Fig. are impossible to read as it stands. Maybe representation as Pie Charts, or other type of graphics allows for larger descriptions.

Fig. 5 Selected 7 DEGs related to flavonoid biosynthesis and their FPKM.
Change to “seven selected differential expressed genes from the transcriptome. Average FKPM values for biological replicates are shown…”.

Fig. 7: Spell out Hydroxysafflor yellow A then display abbreviation.


Table 1 can be moved to Supplementals

Table 2 Please cite the information on the version of the different databases KEGG, SwissProt, and GO as a Figure foot.

Reviewer 2 ·

Basic reporting

English must be improved.

L29: The previous version said that 13 DEGs were found to be related with flavonoid biosynthesis. However now are only 7. I think authors have focused their attention in 7, but 13 is the original number. Am I right? In any case, the result is 13 DEGs found, and this should be said not only in the abstract, also in the results section.

Introduction has been substantially improved in terms of background information. However I have some comments. I suggest some restructuring of phrases and paragraphs. Since first paragraph it is an introduction to safflower crop, I suggest to maybe finish it with the explanation of its distribution with something like this: “Previous studies on the effect of environmental factors on safflower distribution, showed that light was the most important factor explaining safflower distribution boundaries (Wu, 2017)”. And then I would start the second paragraph with the effects of light on plants (If this comment is considered, then phrases 53-59 should be deleted).

Phrases 59-66 should be better connected. The same words are repeated too frequently.

Phrases 64-65 “Numerous studies have shown that light intensity can influence the biosynthesis of flavonoid in other species (Deng et al., 2012; Pan et al., 2017;Wang et al., 2018)”. Describe how is this influence. Does light increase or decrease (or both) flavonoid biosynthesis? Maybe a review about this topic should be cited.

L80: “only a few genes have been identified in flavonoid biosynthesis pathway of safflower”. Describe somehow which genes have been identified.

L167: “Relative DEGs expression analysis” is not a very descriptive and clear title. Maybe something like “RT-PCR expression analysis of DEGs” is more clear for the reader.

L252-253: There is a relative correlation between kaempferol concentration and CtHCT3 expression, ok, but is there any other significant relationship? What about FLS expression and kaempferol concentration? If there are no other significant relationships this should be mentioned.

L259-263: Related with the number of genes identified with flavonoid biosynthesis, in the discussion authors said that other transcriptomes found 156, 22, and 44 unigenes or isoforms related with flavonoid biosynthesis. In this study 13 DEGs were found, however this comparison is not very useful because in this case only the DEGs have been reported and analysed. Do you have an estimation of how many genes related with flavonoid biosynthesis there are in this experiment not only the DEGs? If this information is available it would be nice to put it.

Figure 5: Title of the figure should be something like: “Expression levels of the DEGs related to flavonoid biosynthesis”. Although for the RT-PCR you have selected 7 DEGs, I think that this graphic should include the complete information, that is, the ecpression levels of the 13 DEGs.
Figure 6: Title should also be more descriptive: “Expression analysis by RT-PCR of seven flavonoid related genes under different light intensities”. Be consistent with the fill of the bars across the manuscript. Figure 5 and 6 have different colors for conditions HL and LL. As previously suggested, since B and C are results from HPLC and specific flavonoids, I suggest to put them together instead of A and B. This way the reader can easily appreciate the difference in flavonoid concentration. In the description of the figure, instead of “ANOVA- Tukey test was used for this analysis . Some of the differences were marked in the figure (*P<0.05,**P<0.01)” something like “Results of ANOVA-Tukey tests are shown with asterisks (*P<0.05,**P<0.01)” sounds better.

Experimental design

No comment

Validity of the findings

No comment

Additional comments

MINOR COMMENTS

L17: ingredients → compounds
L17: There is a rare symbol in the word “effects”.
L23: by the DESeq2 method → With the DESeq2 package
L25-26: “Flavonoids accumulation…. by HPLC” → “Flavonoids accumulation…. by HPLC and spectrophotometer”
L27: Light intensity → light intensities
L35: confirmed → suggest
L61: responds →respond
L67: Flavonoid is ubiquitous → Flavonoids are ubiquitous
L69: very clear → well known
L70: p-coumaroyl → P-coumaroyl
L71: Cite Figure 1 after “… phenylpropanoid compounds”.
L78: At the downstream → Downstream
L83: that the flavonoid in safflower → that flavonoids in safflower
L86: Hydroxysafflor yellow A → Specifically, Hydroxysafflor yellow A
L92: ingredient → compound.
L92: Does exist a reference for Chinese Pharmacopoeia? If so, cite.
L143: Delete “After sequencing and read filtering”. Start the sentence with “Clean reads in each sample….”
L147: redundant→ redundant transcripts
L153-154: Move this paragraph to the end of this section
L221: were → are
L276: safflower growth better → safflower growth was better
L276: reached high level → reached the highest level
L296: might → might be
L314: Seven → Thirteen
L316: and their expressions under.. → and the expression of seven of them under..

---

## Round 0.3 · Minor Revisions

Dear authors

As you can see Reviewer 1 is still not happy. You must pay more attention to details.

Kind regards
M. Wink
Academic Editor

Reviewer 1 ·

Basic reporting

Professional english is used. Some minor grammatical errors should be corrected.
For instance, in the last phrase of the introduction: “ In this study, we aimed to explore the relationship between different light intensities and flavonoid biosynthesis in safflower flowers which cultivated in greenhouse.” should be: " we aimed to explore the relationship between different light intensities and flavonoid biosynthesis in safflower flowers cultivated in the greenhouse."

I still miss an important reference of an article published this year on flavonoid biosynthesis in safflower: D. Guo et al. BMC Plant Biol. 2019 Aug 27;19(1):376. doi: 10.1186/s12870-019-1962-0. They discuss the role of CHI an enzyme upstream of the studied enzymes in this paper. It would be nice to see it referred to and possibly show the expression of at least CtCHI1, a nice control to their data.

Figures have captions labeled X- and Y- axis, this is not really standard. Normally the depicted relationships should be described.

The results partially support their hypotheses.

Experimental design

The research is original.

Research question was well defined.

Technical standard needs to be improved, it is difficult to see which is the control in the experiment- as you cannot grow a plant in absence of light. But maybe a "standard" growth from the literature?

They also didn't mention the sample size in each experimental group. Please take care of this.

Also, they didn't show the calculation of primer efficiencies and their answer to the critique point of the reviewer is unsatisfactory. They should show this "95%-105% efficiency" they claim to have in the supplementals.
Their response: "...The method you said should be more accurate, but when we did this part experiment, we generally used the easy-to-use 2 - ΔΔ CT method. Because the amplification efficiency of the target gene and the reference gene was preliminarily determined. The amplification efficiency of target gene and reference gene was close to 100%, and the efficiency deviation between them was within 5%. So we could achieve our goal by using 2 – ΔΔ CT method."

The statistical test is called “ANOVA” with a post-hoc “Tukey- Test” and not ANOVA-Tukey as they call it in the paper. Please check that all assumptions are met, the Tukey test assumes that:
- Observations are independent within and among groups.
- The groups for each mean in the test are normally distributed.
- There is equal within-group variance across the groups associated with each mean in the test (homogeneity of variance).

The other methods seem accurate.

Validity of the findings

Correcting and regarding the technical details mentiones above will improve the paper. The underlying missing data should be provided (sample sizes, primer efficiencies, discussion over controls)

In the discussion (L.272-273) they state: "However, 104 Unigenes were related with flavonoid biosynthesis in this experiment and 13 DEGs related with flavonoids biosynthesis were not found in all above studies." The other studies were not on light intensities, so maybe the statements should relate to the novel results due to the experimental conditions.

Conclusions are well stated.

Additional comments

Dear authors,

The submitted paper “Transcriptome analysis of flavonoid accumulation in safflower flowers grown under different light intensities” is an transcriptome investigation of the flavonoid biosynthesis genes grown under the mentioned conditions. It is a novel aspect for Carthamus tinctorius and is therefore of interest to explore.

Most of the review points I stated above.

In addition to the comments above, please correct the conclusions (CORRECTIONS IN CAPITAL LETTERS):
13 NOVEL genes related to flavonoid biosynthesis were identified based on the functional annotation of differentially expressed genes, and the DIFFERENTIAL expressions of 7 of them under different light intensities were analySed by QRT-PCR. The concentrations of total flavonoid and two active compounds were measured in safflower flowers grown under 3 different LIGHT conditions.

These corrections will lead to a paper in publishing quality.

---

## Round 0.4 · Major Revisions

Dear authors

Our Section editor has read ypur ms and has the following issues:

"I agree that the study had a contribution to the knowledge of flavanoid expression, relating an optimal range of light expression in the production of the flavanoid classes; however, for the highlight of gene classes expressed there is not attachment of the actual sequences. It is especially important that sequence data be provided as the assemblies here are not tied to a reference genome.

There may be opportunity to highlight differences from the reference genome, but to have no sequence data made available does not serve the reader in advancing their knowledge to understand the genes involved. There was considerable discussion of adding value to the transcriptome by way of annotation methods, yet none of the annotations can be tied to actual sequence data.

There needs to be supplementary data which contains the links between the annotations (using the GO: terms) and the sequence data, and perhaps other annotation methods used which were discussed, but not included.

This manuscript serves as a progress report without data, and thus should be placed in a major revision mode until such data can materialize. Journal manuscripts are often scanned by text-mining software that locates and extracts core data elements, like gene function. Adding standard ontology terms, such as the Gene Ontology (GO, geneontology.org) or others from the OBO foundry (obofoundry.org) can enhance the recognition of your contribution and description. This will also make human curation of literature easier and more accurate. None of this was visible. If supplemental data were included, the manuscript is in fine shape for acceptance."

Please address these issues carefully

Regards
Michael Wink
AE

---

## Round 0.5 · accepted · Accept

Dear authors

As the revision is adequate, I am happy to accept your ms.

Congratulations
Kind regards
M. Wink
AE